# A Systematic Bibliometric Review of Low Impact Development Research Articles

Jin You [1,2], Xiang Chen [3,4], Liang Chen [3,4], Jianghai Chen [3,4], Beibei Chai [1,2,*], Aiqing Kang [5], Xiaohui Lei [5] and Shuqian Wang [1]

1. School of Water Conservancy and Hydroelectric Power, Hebei University of Engineering, Handan 056021, China
2. Hebei Key Laboratory of Intelligent Water Conservancy, Hebei University of Engineering, Handan 056038, China
3. Shanghai Investigation, Design and Research Institute Co., Ltd., Shanghai 200434, China
4. China Three Gorges Corporation, Wuhan 430010, China
5. State Key Laboratory of Simulation and Regulation of Water Cycle in River Basin, China Institute of Water Resources and Hydropower Research, Beijing 100038, China
* Correspondence: cbb21@163.com; Tel.: +86-153-4926-8338

**Abstract:** The concept of low impact development (LID) plays a crucial role in rainstorm management and non-point source pollution prevention and control. Sorting and summarizing related research through the knowledge map can objectively present the disciplinary structure, research focus, and research hotspots of the LID research. Based on 2103 LID pieces of literature in Chinese and English included in the web of science (WOS) database and China's integrated knowledge resources system (CNKI) database from 2004 to 2021, this paper aims to perform statistical analysis from three aspects: bibliometrics, keyword hotspot co-occurrence and clustering, and literature co-citation clustering. The obtained results reveal that research on LID-based issues maintains a high degree of enthusiasm in China and abroad, but their corresponding focuses are dissimilar. Foreign research essentially focuses on the environmental field with frequent interdisciplinary phenomena, combining the triple goals of water quality improvement, runoff reduction, and multi-functional expansion, and is committed to solving the impact of uncertain factors on urban stormwater management in extreme climates. Chinese research is mostly aimed at unlocking practical engineering problems, which also leads to the majority of research works in the field of building science and engineering. This is mainly due to a series of water-related problems caused by the change in land use types in China. The researchers have determined the type, quantity, location, and combination of the optimal LID measures by establishing appropriate models, using optimization algorithms, and developing multi-level analysis methods. Although the multi-dimensional results of LID in recent years have greatly expanded the framework paradigm, most of the conducted research works are still biased towards the micro-scale. The present hotspot research considers how to make a macroscopic overall layout and efficiently cooperate with the pipelines network, rivers, and lakes systems to unlock the problems pertinent to urban rainwater and non-point source pollution.

**Keywords:** low impact development (LID); visual analysis; CiteSpace software; research hotspots

## 1. Introduction

Urbanization, industrialization, and climate change significantly affect the intensity, incidence, and duration of extreme weather events, such as floods, urban waterlogging, and nonpoint source pollution [1]. Traditional stormwater management involves the quick discharge of excess stormwater through a centralized drainage system to ensure urban water security [2]. However, the management approach is unsustainable and brings the challenges of groundwater resource depletion, unnatural redistribution of water resources, and excessive carbon emissions [3]. In view of the increasingly severe ecological and

environmental problems, the low impact development (LID) [4,5] design concept based on rain gardens was first proposed in the United States, and relative technical standards was compiled in 1999. Subsequently, various urban stormwater management concepts have been adopted, such as Australia's Water Sensitive Urban Design (WSUD) [6], Sustainable Urban Discharge System (SUDS) [7] in Europe, and China's Sponge City [8,9]. These stormwater management methodologies are aimed at achieving a healthy urban water cycle by coordinating the LID stormwater system, urban stormwater pipes and canal system, and the excess stormwater runoff discharge system. The LID is an essential part of stormwater management that emphasizes green priority and source control. Through green LID measures such as bioretention, grass trenches, and green roofs, urban stormwater can be managed from the source via infiltration, storage, regulation, transfer, and pollutant interception and purification [10]. By increasing investigations on the LID, more and more results have proved that LID lessens the total amount of runoff [11,12] and peak flood flow [13], delays the peak flood time [14], rises the infiltration rate [15], and reduce the pollution load [16]. However, it leads to obtaining synergistic benefits in terms of water environment and ecology [17].

By collecting LID-based articles published from 2004 to 2021 in Chinese and English in accordance with the WOS and CNKI databases, the information visualization analysis was performed using appropriate software entitled "CiteSpace". The main objective of exploiting this specific software is to conduct statistical analysis and visual map demonstration from various aspects. The most important ones include bibliometrics, hotspot keyword co-occurrence and clustering, and co-citation clustering, in an effort to systematically review the current LID research. While exploring the LID research by international scholars, the research status in China is also reviewed. In the context of the sponge city construction and sustainable development in China, the research is aimed at providing rational solutions for some crucial issues such as urban waterlogging, groundwater resource depletion, and urban nonpoint source pollution.

## 2. Method and Data Processing

In recent years, the bibliometric literature review has been broadly performed in hydrology [1,18], ecology [19], regenerative medicine [20], and other fields. Using visualization software, one can directly, clearly, and vividly diagnose the dynamic evolution, development trend, research progress, and hotspots in the research field. CiteSpace [21,22] is extensively utilized in analyzing keywords and citations of articles due to its powerful function, excellent visualization effects, and rich layouts. Keywords are direct descriptions of the content and theme of a study, and citations comprehensively display the study details and the development trend. In the keyword co-occurrence analysis, the frequency of keywords usually reveals the evolution of keyword hotspots in the field of research and the strength of individual keywords. In combination with burst terms, it can review the volatile hotspots in the research field and explore the potential applications. Keyword cluster analysis is to classify data into various groups according to the homogeneity between keywords to obtain high support directions of different categories. Co-citation analysis can be utilized to show the research content of key nodes, such that it is convenient for researchers to summarize current research clusters and analyze future research directions.

In the present scrutiny, a total of 2103 articles in the field of LID research are collected. Of which, 1053 articles written in English are retrieved from the Web of Science (WOS) database. The timeframe is between 2004 and September 2021. The keyword used in the literature search is low impact development. The literature type is limited to articles. The CNKI database is also exploited for Chinese literature retrieval, the understudy timeframe is between 2009 and September 2021, the keyword used for search is "low impact development + LID", and the literature type includes both journal publications and dissertations, and a total of 1050 articles are collected. Then, the CiteSpace5.6.R5 software (CiteSpace 5.6.R5, Chaomei Chen: Philadelphia, PA, USA) is employed to analyze the quantity, subject categories, keywords, and co-citations of the included articles. This software can summa-

rize the research status, progress, and hotspots, as well as the associated problems with those studies. Finally, research prospects are proposed and explained in some detail.

## 3. Bibliometric Analysis of LID-Based Research Articles

Based on the bibliometric analysis of the data derived from WOS and CNKI, the variation of the annual number of publications and the cumulative number is demonstrated in Figure 1. It can be seen from Figure 1 that the cumulative numbers of articles written in Chinese and English are generally comparable. The LID research abroad emerged in 2004, and then the annual number of articles has steadily grown. The Chinese articles on the LID originated in 2009, and the number of articles rapidly increased after 2015. The concept of LID is originally proposed by the Prince George's County Department of Environmental Resources in the US. Subsequently, on the basis of the LID concept, several other concepts are proposed globally, such as Australia's Water Sensitive Urban Design (WSUD), New Zealand's Low Impact Urban Design and Development (LIUDD) program, Sustainable Urban Drainage System (SUDS) in Europe and the Sponge Cities in China. In Graham et al. [23] published in 2004, the specific concept of "Low Impact Development" was employed for the first time, and the efficiency of the LID measures such as bioretention, green roofs, and permeable pavement was assessed through water balance simulation. In China, the construction of pilot sponge cities in 2015 can be regarded as a key time point in the LID research.

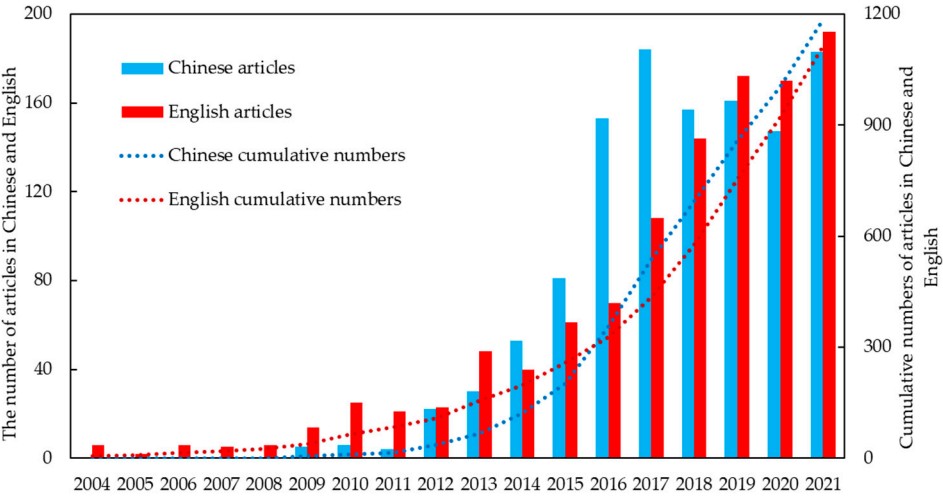

**Figure 1.** The number of published papers on the LID framework research from 2004 to 2021.

From the subject categories of the WOS literature as presented in Table 1, the largest categories of English articles are Environmental Sciences & Ecology, Environmental Sciences, and Water Resources. In these English-based studies, the runoff and pollution caused by heavy rainfall have been managed through source control such that the development area has a natural hydrological cycle as much as possible. Islam et al. [1] proposed the overall goals of the LID, which includes: (1) comprehensive control of the precipitation and water quality, (2) control of the runoff from the source as much as possible, (3) increasing urban hydrological processes, such as evaporation, infiltration, and storage, and (4) improving ecological benefits. In addition to the theoretically performed research, the application of LID in various fields, including engineering, civil engineering, and engineering and environment, has been examined by abroad researchers.

**Table 1.** The subject categories of the LID research based on the WOS database.

| No. | Quantity | Centrality | Subject |
|---|---|---|---|
| 1 | 689 | 0.35 | Environmental Sciences & Ecology |
| 2 | 637 | 0.11 | Environmental Sciences |
| 3 | 560 | 0.08 | Water Resources |
| 4 | 467 | 0.53 | Engineering |
| 5 | 232 | 0.16 | Engineering & Civil |
| 6 | 212 | 0.03 | Engineering & Environmental |

Different from of the research abroad, the largest categories of Chinese-based studies include Construction Science & Engineering and Water Conservancy & Hydropower Engineering (see Table 2). This is mainly attributed to the fact that the rapid urbanization in China is the primary challenge for urban stormwater management. The increase in urban construction land has changed rivers outside the city into inland rivers, reducing stormwater storage space, increasing impervious areas, and aggravating urban waterlogging. Moreover, urban construction has a huge impact on the ecological environment and exerts great pressure on water security. With the construction of the pilot sponge cities in 2015, Chinese scholars have conducted extensive examinations on the difficulties and demands in the construction of sponge cities, which greatly expand the application field of the LID research.

**Table 2.** The subject categories of the LID research based on the CNKI database.

| No. | Quantity | Centrality | Subject |
|---|---|---|---|
| 1 | 919 | 61.27 | Construction Science & Engineering |
| 2 | 357 | 23.80 | Water Conservancy & Hydropower Engineering |
| 3 | 88 | 5.87 | Environmental Science and Resource Utilization |
| 4 | 72 | 4.80 | Road and Water Transportation |
| 5 | 8 | 0.53 | Resource Science |
| 6 | 4 | 0.27 | Gardening |

## 4. International LID Research Progress

### 4.1. Keyword Co-Occurrence Analysis of Research Topics

The LID is a novel stormwater management method, and its whole evolution can be divided into the formation and development stages. According to the statistics of high-frequency keywords (see Table 3), the formation stage can be rationally considered in the time interval of 2004–2014. In this stage, the context of the LID research is finalized, and the impact on the hydrological cycle, model simulation, and practical application are discussed. The top keywords include Performance, System, as well as those representing crucial components of the hydrological cycle such as Runoff, Infiltration, and Water Quality. The SWMM is broadly utilized in the LID research due to its excellent hydrological and hydrodynamic simulation performance. In addition, Bioretention and Green Infrastructure are also repeatedly mentioned as potential application directions. The development stage of the LID started in 2015. In this stage, the focus is on the Optimization, Uncertainty, and Resilience along with Management Practice. The research content involves perfecting the LID research framework, as well as addressing new demands such as water security and water ecology brought about by global climate change.

**Table 3.** The high-frequency keywords of publications according to the WOS database.

| Stage | Keywords (Frequency/Centrality) |
|---|---|
| Formation stage (2004–2014) | LID (503/0.04); runoff (249/0.05); performance (220/0.02); stormwater management (191/0.06); system (153/0.09); urbanization (140/0.07); bioretention (137/0.06); green infrastructure (133/0.04); water quality (113/0.04); model (107/0.03); SWMM (105/0.03); infiltration (79/0.08) |
| Development stage (2015–2021) | optimization (68/0.01); management practice (63/0.01); infrastructure (31/0.01); city (30/0.02); urban runoff (28/0.01); uncertainty (25/0.04); benefit (22/0.01); resilience (17/0.01); reduction (17/0.01) |

It can be found based on the burst terms in English-based articles (see Table 4, red colors represent the year of the burst terms appeared) that Stormwater Management has been a research hotspot throughout the research period due to its central role in the LID-based research. The LID measures such as Porous Pavement and Bioretention emerged as short-term research hot topics. In recent years, the hotspots have shifted to Stormwater Management Model and Challenges due to global climate change.

**Table 4.** Keywords burst detection of publications based on the WOS database.

| Keywords | Year | Strength | Begin | End | 2004–2021 |
|---|---|---|---|---|---|
| best management practice | 2004 | 8.5 | 2004 | 2015 | |
| infiltration | 2004 | 7.22 | 2004 | 2015 | |
| storm water management | 2004 | 3.81 | 2004 | 2015 | |
| storm water | 2004 | 6.2 | 2005 | 2016 | |
| porous pavement | 2004 | 6.53 | 2007 | 2017 | |
| retention | 2004 | 4.8 | 2007 | 2016 | |
| quantity | 2004 | 3.82 | 2007 | 2010 | |
| hydrology | 2004 | 4.89 | 2008 | 2013 | |
| stream | 2004 | 4.42 | 2008 | 2017 | |
| water | 2004 | 4.47 | 2009 | 2014 | |
| bioretention | 2004 | 3.79 | 2009 | 2013 | |
| North Carolina | 2004 | 5.6 | 2010 | 2014 | |
| pollutant removal | 2004 | 5.53 | 2011 | 2015 | |
| storm water management | 2004 | 3.94 | 2011 | 2012 | |
| land use change | 2004 | 5.41 | 2015 | 2016 | |
| urban runoff | 2004 | 5.25 | 2015 | 2017 | |
| modeling | 2004 | 4.14 | 2015 | 2017 | |
| rainfall | 2004 | 5.67 | 2019 | 2021 | |
| challenge | 2004 | 4.49 | 2019 | 2021 | |
| storm water management model | 2004 | 3.85 | 2019 | 2021 | |

Notes: Red colors represent the year of the burst terms appeared.

### 4.2. Keyword Cluster Analysis

According to the keyword clustering (see Table 5) and the timeline view of clustering (see Figure 2), it is detectable that the research on Bioretention (#0) is relatively independent, and the experimental analysis and water quality improvement (contaminant removal rate) have been the focused of the clustering study. In contrast, the other LID measures such as Green Roof (#2) and Green Infrastructure (#4) focused more on the model simulation and water reduction (e.g., total runoff, peak flow, and peak flood time). Additionally, landscape ecological benefit is a performance indicator of all LID measures. Liu et al. [24] conducted a detailed investigation on the principle, design, performance, advantages, disadvantages, and costs of LID measures such as bioretention ponds, grass-planting trenches, sunken green spaces, green roofs, permeable pavements, rainwater tanks, and roof truncations.

Optimization (#3) and Climate Change (#7) are the focus of the current research, the clustering keywords such as Optimization, Uncertainty, Cost-effectiveness, Climate change, and Urbanization represent the hot topics and the leading edge of the current research. In the cluster, the cost has been considered as the objective function in some articles, and they are aimed to scrutinize the distribution strategy for reducing the total runoff, peak runoff, and pollutant load. With more and more in-depth investigations, some scholars have commenced considering the influences of externally uncertain conditions such as climate change, urbanization, and rainfall. Furthermore, the parameter optimization of the LID measures (e.g., NSGA II, PSO, MOALOA) and multi-criteria methods [25] (e.g., AHP, TOPSIS) has attracted much attention. Gogate et al. [26] employed AHP and TOPSIS multi-criteria analysis approaches to choose the best storm control measures. Li et al. [27] discovered the optimal combination of LID measures and retention tanks by combining the SWMM model with the PSO algorithm. The SWMM is the most common model exploited in the LID research. Kaykhosravi et al. [28] conducted a comprehensive analysis of eleven LID models as a function of the model characteristics, hydrology, and hydraulic modules and found that the SWMM had the best applicability.

**Table 5.** The keyword clustering of the publications based on the WOS database.

| Cluster Name | Size | Homogeneity | Research Topic (log Likelihood Ratio/$p$ Value) |
|---|---|---|---|
| #0 bioretention | 112 | 0.616 | bioretention (48.76/0.001); SWMM (31.51/0.001); stormwater (30.66/0.0001) |
| #1 sponge city | 106 | 0.457 | sponge city (38.37/0.0001); bioretention (18.038/0.001); urban catchment (13.75/0.001); water quality (13.68/0.001); sustainablity (13.19/0.001) |
| #2 green roof | 69 | 0.669 | green roof (28.95/0.0001); porous pavements (21.32/0.0001); permeable pavement (21.04/0.0001) |
| #3 optimization | 61 | 0.694 | optimization (37.09/0.0001); cost (17.07/0.001); management practice (16.46/0.0001); uncertainty (11.33/0.001) |
| #4 green infrastructure | 57 | 0.697 | green infrastructure (24.35/0.0001); stormwater management (20.31/0.001); urban hydrology (17.28/0.0001) |
| #5 low impace development | 36 | 0.745 | low impact development (32.02/0.0001); SWMM model (28/0.0001); LID (12.97/0.001) |
| #6 constant head test | 34 | 0.829 | constant head test (12/0.001); pollution (12/0.001); sensitivity analysis (10/0.005) |
| #7 climate change | 33 | 0.784 | climate change (35.15/0.0001); urbanization (31.45/0.05); land use (13.78/0.001) |
| #8 SWMM | 28 | 0.873 | SWMM (44.37/0.0001); water quality (10.5/0.005); conservation subdivision (10.17/0.005) |

*4.3. Co-Citation Clustering Analysis*

Based on the co-citation clustering (see Table 6) and the co-citation references clustering network (see Figure 3), it can be seen that the LID research can be divided into the three major groups: optimization (#0 multi-objective optimization), principle (#1 hydrology, #2 groundwater, and #6 first flush effect), and application (#3 sponge city, #4 LID practices, #5 green roof). It can be seen from the clustering timeline (see Figure 4) that the research on the principle of the LID originated in the field of hydrology. Since 2004, a large number of investigations have been carried out on the impact of the LID measures on the hydrological processes, which demonstrate that precipitation is the main factor that affects the hydrological process of the LID measures. Qin et al. [29] systematically examined extreme rainfall events with various precipitation, duration, and peak intensity, and analyzed the performance of the LID measures due to different precipitation characteristics. Since 2012, urban hydrology and groundwater recharge have become the research hotspots. Palla and Gnecco [10] conducted a simulation examination of the LID-based systems on the urban watershed scale and validated the effectiveness of the LID measures in various rainfall return periods. In the past few years, the research on the principle of the LID

measures mostly focused on the initial scour effect, which is also the main reason for the urban nonpoint source pollution. Yang et al. [30] proposed an appropriate classification standard for multi-objective stormwater management, which is employed to select suitable bioretention facilities considering multiple performance objectives (e.g., reduction of initial scour effect, and lessening of the total runoff and peak flow).

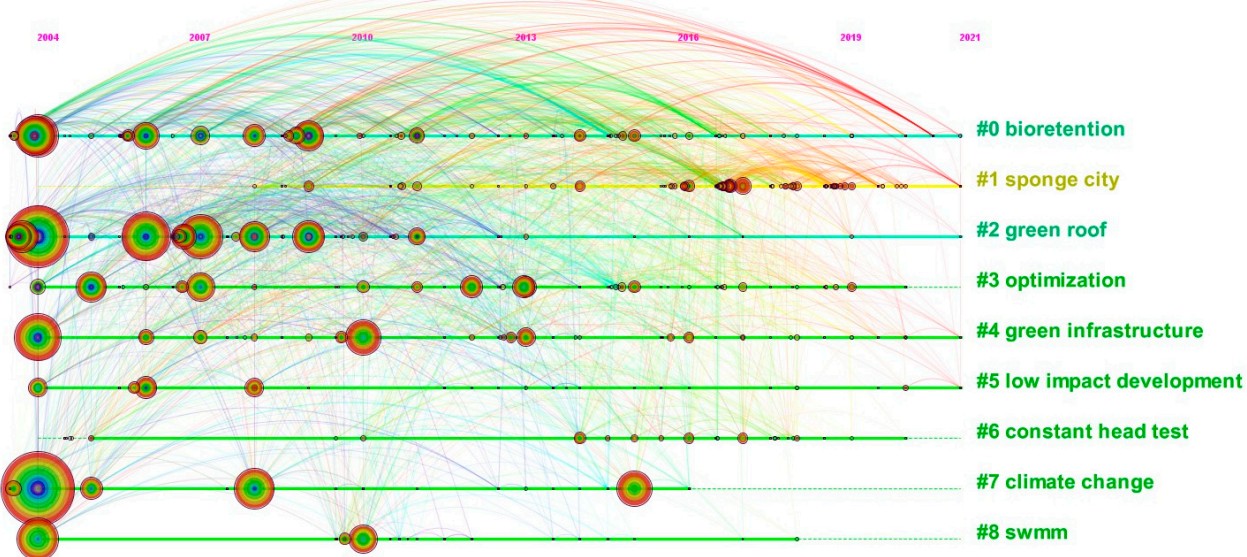

**Figure 2.** The time-line view of the keywords clustering based on the WOS.

**Table 6.** The co-citation clustering of the LID research according to the WOS.

| Cluster | Size | Homogeneity | Average Year | Research Topic (log Likelihood Ratio/*p* Value) |
|---|---|---|---|---|
| #0 multi-objective optimization | 163 | 0.717 | 2017 | multi-objective optimization (11.77/0.001); optimization (10.98/0.001); SWMM (8.97/0.005); climate change (6.33/0.05); sustainable urban drainage systems (6.02/0.05); life cycle cost (5.77/0.05) |
| #1 hydrology | 132 | 0.902 | 2008 | hydrology (22.52/0.0001); bioretention (14.78/0.001); SWMM (12.47/0.001); sponge city (12.23/0.001); sustainable development (12.06/0.001) |
| #2 groundwater | 85 | 0.781 | 2014 | groundwater (9.36/0.005); urban hydrology (9.08/0.005); green infrastructure (12.47/0.001); groundwater recharge (7.95/0.005) |
| #3 sponge city | 71 | 0.796 | 2016 | sponge city (25.75/0.0001); bioretention (12.92/0.001); urban sustainability (7.81/0.01) |
| #4 LID practices | 57 | 0.875 | 2011 | LID practices (6.77/0.01) |
| #5 green roof | 52 | 0.897 | 2013 | green roof (26.31/0.0001); LID (12.2/0.001); retention (10.79/0.005) |
| #6 first flush effect | 46 | 0.95 | 2017 | first flush effect (10.22/0.0005); denitrification (10.22/0.005); stormwater runoff (8.93/0.005) |
| #7 sustain | 30 | 0.921 | 2011 | sustain (6.93/0.01); site selection (6.13/0.05); LID-BMPs (6.13/0.05) |
| #8 conservation subdivision | 26 | 0.979 | 2007 | conservation subdivision (16.32/0.0001); experimental auction (8.13/0.005) |
| #10 benefit cost ratios | 21 | 0.952 | 2002 | benefit cost ratios (10.43/0.005); conservation (10.43/0.005); water balance (7.67/0.01) |

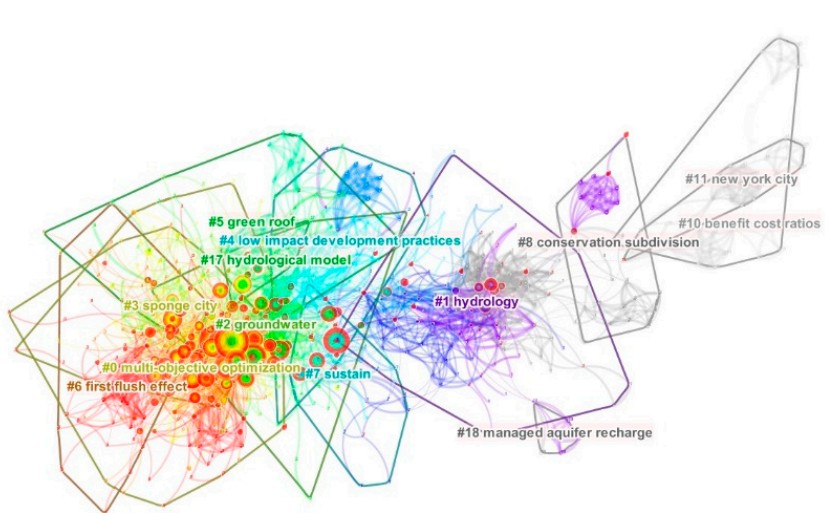

**Figure 3.** The co-citation references clustering network of the LID framework research based on the WOS.

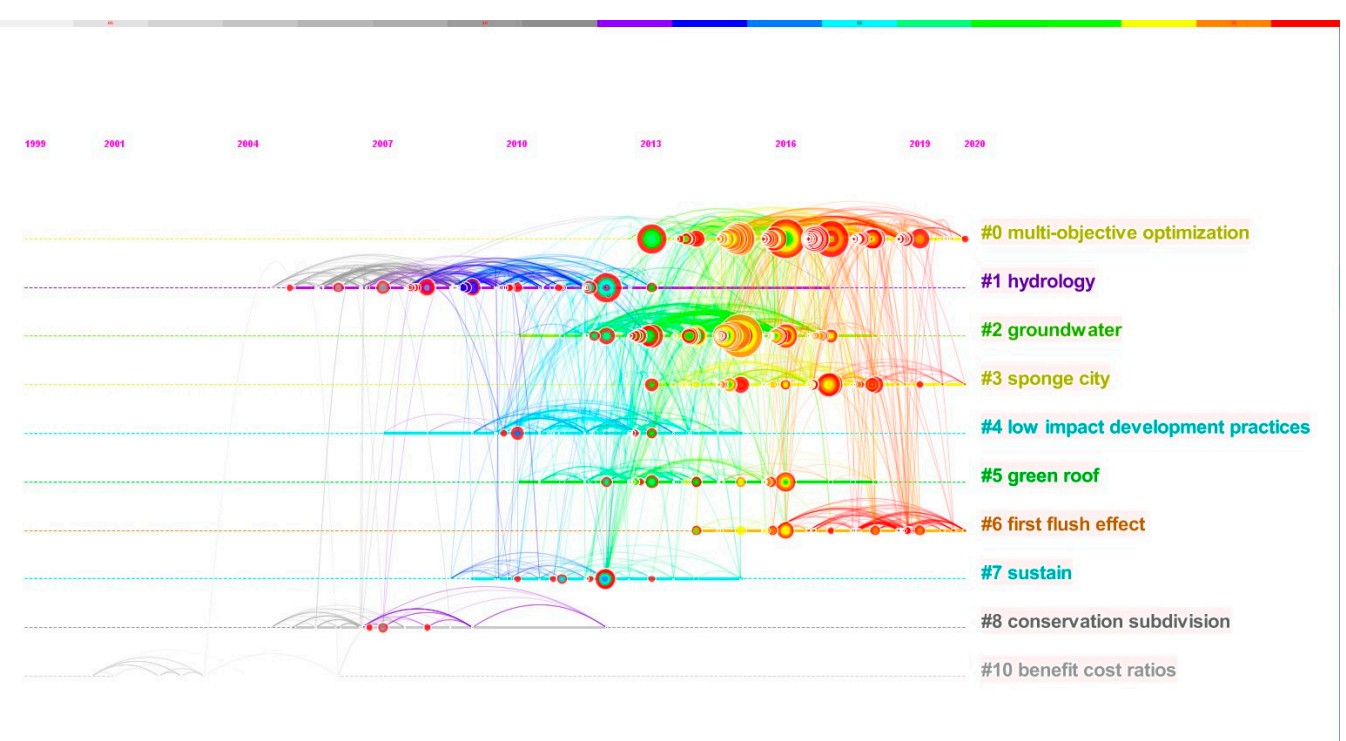

**Figure 4.** The time-line view of the LID framework research based on the WOS.

Table 7 presents high-cited literatures in the field of low-impact development re-search. From the perspective of literature citations, scholars in other countries have examined green infrastructure and bioretention in the early stage. However, after 2015, the performed research works have been mainly focused on the construction of sponge cities in China. For instance, Xia et al. [31] and Li et al. [32] summarized the opportunities and challenges facing the construction of sponge cities in China. LID optimization has been a hot topic since it was first mentioned in 2013. The relevant studies adopted the SWMM and optimization algorithm to solve the objective function (e.g., water quantity [33,34], water quality [35], and

cost minimization [36], thereby determining the optimal deployment of the LID measures. Some other explorations also considered uncertainties in the input parameters such as precipitation, anti-seepage coefficients, accumulation and scour coefficients, as well as changes brought about by climate change [37] and land-use type [38].

**Table 7.** The key publications from co-citation references clustering according to the WOS.

| Frequency | Year | Author | Title |
|---|---|---|---|
| 122 | 2017 | Eckart et al. (2017) [8] | Performance and implementation of low impact development—A review |
| 109 | 2015 | Fletcher et al. (2015) [9] | SUDS, LID, BMPs, WSUD and more—The evolution and application of terminology surrounding urban drainage |
| 79 | 2015 | Palla and Gnecco (2015) [10] | Hydrologic modeling of Low Impact Development systems at the urban catchment scale |
| 75 | 2016 | Ahiablame and Shakya (2016) [13] | Modeling flood reduction effects of low impact development at a watershed scale |
| 75 | 2016 | Chui et al. (2016) [36] | Assessing cost-effectiveness of specific LID practice designs in response to large storm events |
| 64 | 2013 | Qin et al. (2013) [29] | The effects of low impact development on urban flooding under different rainfall characteristics |
| 61 | 2012 | Ahiablame et al. (2012) [5] | Effectiveness of low impact development practices: literature review and suggestions for future research |
| 58 | 2015 | Baek et al. (2015) [33] | Optimizing low impact development (LID) for stormwater runoff treatment in urban area, Korea: Experimental and modeling approach |
| 51 | 2015 | Liu et al. (2015a) [34] | Enhancing a rainfall-runoff model to assess the impacts of BMPs and LID practices on storm runoff |
| 50 | 2015 | Rosa et al. (2015) [39] | Calibration and verification of SWMM for low impact development |
| 49 | 2017 | Kong et al. (2017) [38] | Modeling stormwater management at the city district level in response to changes in land use and low impact development |
| 47 | 2015 | Jia et al. (2015) [40] | LID-BMPs planning for urban runoff control and the case study in China |
| 46 | 2019 | Li et al. (2019) [41] | Comprehensive performance evaluation of LID practices for the sponge city construction: A case study in Guangxi, China |
| 45 | 2018 | Eckart et al. (2018) [42] | Multiobjective optimization of low impact development stormwater controls |
| 44 | 2017 | Xia et al. (2017) [31] | Opportunities and challenges of the sponge city construction related to urban water issues in China |
| 41 | 2015 | Martin-Mikle et al. (2015) [43] | Identifying priority sites for low impact development (LID) in a mixed-use watershed |
| 40 | 2017 | Mao et al. (2017) [44] | Assessing the ecological benefits of aggregate LID-BMPs through modelling |
| 40 | 2017 | Li et al. (2017) [32] | Sponge city construction in China: A survey of the challenges and opportunities |
| 40 | 2018 | Zhang and Chui (2018) [45] | A comprehensive review of spatial allocation of LID-BMP-GI practices: Strategies and optimization tools |
| 39 | 2015 | Liu et al. (2015b) [35] | Evaluating the effectiveness of management practices on hydrology and water quality at watershed scale with a rainfall-runoff model |

## 5. LID Research Progress in China

### 5.1. Keyword Co-Occurrence Analysis of Research Topics

According to the high-frequency keywords listed in Table 8, the research contents of Chinese articles in the formation stage (i.e., the time interval of 2009–2014) are basically the same as those of English articles, mainly focusing on the nature of LID and sponge cities, as well as the applications of both the SWMM and SUSTAIN models. In the development stage (i.e., the time interval of 2015-present), different from research works abroad that have focused on global climate change, China's research works have been mostly focused on solving the negative effects of urbanization, including an in-depth exploration of landscape design and sponge city construction such as rainfall gardens, LID facilities, landscaped

gardens, and sponge campuses. It is should be noticed that at this stage, the total annual runoff control is proposed as the general target of the LID design in China, and the concept of sustainable development of sponge city construction is also defined.

**Table 8.** The high-frequency keywords of publications according to the CNKI database.

| Stage | Keywords (Frequency/Centrality) |
|---|---|
| Formation stage (2009–2014) | LID (756/1.21); Sponge city (336/0.26); SWMM mode (125/0.11); Stormwater management (57/0.03); Rainwater utilization (37/0.01); Cost-effectiveness(31/0.01); Urban waterlogging (29/0.01); Runoff control (274/0.01); Urbanization (24/0.01); Green roof(17/0.01); Non-point source pollution (16/0.00); SUSTAIN model(15/0.00) |
| Development stage (2015–2021) | Rainfall garden (35/0.01); Landscape design (31/0.01); LID facilities(22/0.04); Landscape garden (21/0.01); Rainwater system (20/0.01); total annual runoff control (15/0.00); Sponge campus (13/0.00); Stormwater management model (12/0.00); Sustainable development (11/0.00) |

Based on the burst terms displayed in Table 9, it can be found that present China's LID research is mainly conducted in three branches: rainfall gardens, residential areas, and sponge campuses. The sponge campus is a small urban stormwater system and is a hot topic of current research, whose burst strength reaches 3.3. Additionally, a variety of approaches such as AHP and multi-objective optimization have been exploited to determine the optimal type [46], size, as well as quantity, and location [47] of the LID measures.

**Table 9.** The burst terms of publications according to the CNKI database.

| Keywords | Year | Strength | Begin | End | 2009–2021 |
|---|---|---|---|---|---|
| storm water | 2009 | 3.8 | 2009 | 2015 | |
| cost-effectiveness | 2009 | 3.46 | 2009 | 2014 | |
| runoff | 2009 | 3.42 | 2009 | 2014 | |
| urbanization | 2009 | 3.3 | 2009 | 2015 | |
| green building | 2009 | 2.59 | 2009 | 2016 | |
| optimization algorithm | 2009 | 2.43 | 2009 | 2014 | |
| sponge city | 2009 | 2.12 | 2009 | 2014 | |
| storm-water management | 2009 | 3.81 | 2012 | 2015 | |
| rainwater utilization | 2009 | 3.04 | 2012 | 2015 | |
| rain garden | 2009 | 2.14 | 2014 | 2016 | |
| the amount of runoff | 2009 | 2.1 | 2014 | 2015 | |
| residential area | 2009 | 2.66 | 2016 | 2018 | |
| planning and design | 2009 | 2.39 | 2017 | 2018 | |
| sponge campuses | 2009 | 3.32 | 2019 | 2021 | |
| analytic hierarchy process | 2009 | 2.9 | 2019 | 2021 | |
| multi-objective optimization | 2009 | 2.53 | 2019 | 2021 | |
| low impact development | 2009 | 2.23 | 2019 | 2021 | |
| stormwater management model | 2009 | 2.2 | 2019 | 2021 | |
| urban road | 2009 | 2.08 | 2019 | 2021 | |
| stormwater regulation | 2009 | 2.07 | 2019 | 2021 | |

Notes: Red colors represent the year of the burst terms appeared.

*5.2. Keyword Cluster Analysis*

From the keyword cluster analysis (see Table 10) and the timeline view of the clustering (see Figure 5) standpoints, the demonstrated results reveal that the cluster is relatively simple. The keywords in the Grass trenches (#2), Rainwater Utilization (#3), and Sponge

City (#4) clusters are mostly presenting the application areas. According to Figure 5, the key nodes are concentrated in the early stage. The prominent nodes are concentrated in the early stage of the introduction framework, and there are few connections between clusters, reflecting that domestic LID is mostly based on practical problems and lacks in-depth exploration of theories. Some scholars have examined the influence of external factors such as confluence relationship [48], terrain slope [49], and pre-conditions [50] on the effectiveness of LID measures, and the research results have enriched the scope of LID-based frameworks. Several investigators have also verified the effectiveness of LID-based measures in the established LID research areas such as Beijing [51], Guangxi [41], and Shenzhen [52].

**Table 10.** The keyword clustering of publications based on the CNKI database.

| Cluster | Size | Homogeneity | Research Topic (log Likelihood Ratio/$p$ Value) |
|---|---|---|---|
| #0 LID | 93 | 0.779 | LID (71.82/0.0001); Urban waterlogging (39.9/0.0001); Sustainable development (18.18/0.0001) |
| #1 SWMM model | 70 | 0.719 | SWMM model (69.66/0.0001); Sponge-type utility tunnel (45.16/0.0001); SUSTAIN model (45.16/0.0001) |
| #2 Grass trenches | 53 | 0.841 | Grass trenches (45.54/0.0001); Green roof(35.47/0.0001); permeable pavement (22.55/0.0001) |
| #3 Rainwater utilization | 50 | 0.824 | Rainwater utilization (43.52/0.0001); LID technologies (35.14/0.0001); Landscape design (26.85/0.0001) |
| #4 Sponge city | 47 | 0.697 | Sponge city (93.67/0.0001); Benefit quantification (29.77/0.0001); Optimization (29.77/0.0001); Annual runoff control rate (23.15/0.0001) |

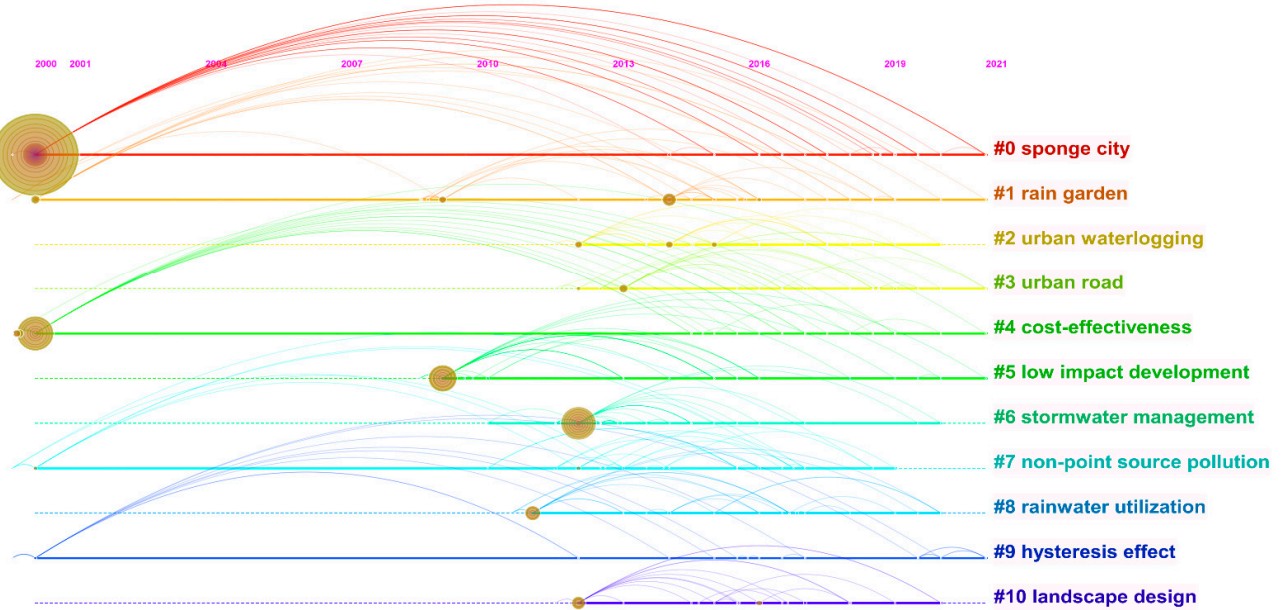

**Figure 5.** The time-line view of the keywords clustering of publications according to the CNKI database.

## 6. Conclusions

### 6.1. Comparison of the LID Research in China and Abroad

Research on LID-based issues has maintained a high level of interest at China and abroad, but their pertinent focuses are different. By comparing the LID research in China and abroad, the following results are obtained:

(1)  Research abroad focuses on the improvement and expansion of the LID framework, while China's research emphasizes on the application of technologies associated with

the sponge city construction. From the number of publications by subject category point of view, the research field with the largest number of publications in English articles is Environmental Science, whereas the top research field in China is Architectural Science. This issue is closely related to the severe water safety as well as water ecological problems in China. Compared to the relatively mature urban stormwater management systems in foreign countries, the rapid urbanization process in China has resulted in the lack of long-term planning for urban stormwater management systems. Since the proposal of sponge city construction in 2015, the number of related Chinese articles has grown exponentially. The conducted studies not only solve the problems of urban waterlogging, groundwater resource depletion, and urban non-point source pollution in China, but also provide references for the LID application in other countries;

(2) Commonly, the research abroad emphasizes on the impact caused by global climate change, whereas China's research is mainly motivated by changes in land-use types. From the keyword clustering standpoints, research abroad basically focuses on the LID parameter optimization, modelling, and improvement of the multi-objective optimization method, uncertainty due to climate and land-use changes, and disaster resistance, while China's research emphasizes on problems encountered in sponge city construction. The advantage is that, in the context of urban waterlogging and non-point source pollution, such explorations can quickly and comprehensively analyze the problems and provide timely information for China's sponge city construction, sustainable development, water safety, and water ecology. Nevertheless, there are also some apparent shortcomings, such as high homogeneity, and lack of innovation;

(3) The research abroad focuses on optimization of the LID parameters through experimental research, while China's research emphasizes on the model simulation in order to determine the type, quantity, location, and combination of optimal LID measures. The bioretention represents the largest cluster of English articles, and Experimental Analysis and Water Quality Improvement are the focus of such investigations. However, few Chinese explorations have been carried out in the areas mentioned above. China's research mainly employs the SWMM model to simulate the LID measures with obvious water reduction effects, such as concave green space and permeable pavement, subjected to various rainfall return periods.

*6.2. Research Trends*

Based on the above discussion, climate change and disaster mitigation are frontier issues in the fields pertinent to the LID. However, the present literature has some limitations, which still should be discussed deeply as follows:

(1) Expand the scope of research. The LID research has achieved good results in research areas such as sponge campuses, residential areas, and urban "brownfields", but most of research works have been conducted at the micro-scale. How to extend the research scale to the scale of cities and watersheds, make a macro-level overall layout, and proficiently cooperate with the pipeline networks, rivers, and lake systems is the key to unlocking the problems of urban rainwater and non-point source pollution;

(2) Expand the breadth of research. To further strengthen the cross-coordination between various disciplines, the previous disciplines were mainly involved in the fields of water resources and the environment, and the optimization objectives and optimization functions were mostly considered as water quality and water quantity. Whether multi-objective optimization can be taken into account in the future, economic benefits, landscape benefits, and climate uncertainties are also included in the evaluation system;

(3) Carry out the original view test. The LID measure can be regarded as a small hydrological cycle system whose efficacy depends on both its corresponding factors and externally exerted conditions. Foreign scholars have accomplished a lot of experimental work on bioretention facilities to assess their performance. Prototype observation

experiments for other LID measures can be also carried out in the future, and the obtained results can be employed to validate the numerical models for the sake of enhancing the research accuracy.

**Author Contributions:** Conceptualization, J.Y. and B.C.; methodology, J.Y., B.C. and S.W.; validation, J.Y., X.C., L.C. and J.C.; data curation, J.Y., B.C., A.K. and X.L.; writing original draft preparation, J.Y.; writing—review and editing, B.C. and S.W. All authors have read and agreed to the published version of the manuscript.

**Funding:** This work was supported by the National Natural Science Foundation of China (NSFC, No. 51809283), and the Foundation of China Three Gorges Corporation (No. 202003136).

**Institutional Review Board Statement:** Not applicable.

**Informed Consent Statement:** Not applicable.

**Data Availability Statement:** The datasets used and/or analysed during the current study are available from the corresponding author on reasonable request.

**Acknowledgments:** The authors would like to express their gratitude to Hebei University of Engineering, and EditSprings (https://www.editsprings.cn, accessed on 18 March 2022) for the expert linguistic services provided.

**Conflicts of Interest:** The authors declare no conflict of interest.

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
