# Peer review of "A Systematic Bibliometric Review of Low Impact Development Research Articles"

_water, doi:10.3390/w14172675_

Round 1

Reviewer 1 Report

Very interesting work although it is simple and direct.

Figure 6 and 7 should be accompanied with English translation. It is not acceptable to leave this part not understandable to most of the readers. Simple translation in brackets will help to give an indication of the meaning.

Author Response

Dear Reviewer:

We would like to thank you for spending so much time reading our manuscript entitled “A systematic bibliometric review of low impact development research articles” (ID: water-1846275) and for providing valuable comments and suggestions. We would like to thank the reviewers for their thoughtful review of this manuscript. The comments and suggestions are very helpful for improving the manuscript. We have revised our manuscript accordingly, and the major revisions are marked in red. Our point-by-point responses are as follows.

第1点图6、图7需附英文翻译。让大多数读者无法理解这部分内容是不可接受的。括号中的简单翻译将有助于说明含义。

我们的回应1:我们一致同意审稿人关于将图6和图7的中文部分替换为英文的意见,这样可以更好地帮助读者理解手稿的含义。在重新创建图 6 和图 7 时,我们同时按照建议将图 6 替换为表 9(第 264 行),并将图 7 替换为图 5(第 279 行)。请看修改后的手稿。

Reviewer 2 Report

In this review paper, the authors collected the low-impact development (LID)-related publications published in Chinese and English based on the WOS (Web of Science) and CNKI (China Integrated Knowledge Resources System) databases from 2004 to 2021 using the software “CiteSpace” to carry out statistical analysis and visualization.

This study is very interesting and can be used as a reference study for readers working on this subject.

However, the manuscript needs to be checked before being accepted.

Additionally, please consider the following points to improve the manuscript:

  Page 1_Line 34: I think the citation must be numbered!!! Please check the journal requirements for the citation & references!!!

  Move figure information after the figures!!!

  Regarding Figure 1: No need to write “published”. Remove it

  And, include “in Chinese and English” after “The number of articles”...

  Include “of articles in Chinese and English” after “cumulative numbers”…

  Close to the graph.

  It is interesting to note that the number of Chinese articles increased between 2014 and 2018 compared to English articles. Then, this trend started decreasing for the Chinese. What was the reason for that? Explanation...

  No explanation for Table 6 in the text!!!

  Where is Table 6 in the text???

  Figure 6: Please translate “Chinese words” to English in Figure 6.

  Finally, there are some mistakes in the writing of the paper.

  Please check the attached file for my corrections/comments.

Besides this, the manuscript can be accepted after the major revision, then, it can be considered to be published in the journal of Water.

Author Response

Dear Reviewer:

We would like to thank you for spending so much time reading our manuscript entitled “A systematic bibliometric review of low impact development research articles” (ID: water-1846275) and for providing valuable comments and suggestions. We would like to thank the reviewers for their thoughtful review of this manuscript. The comments and suggestions are very helpful for improving the manuscript. We have revised our manuscript accordingly, and the major revisions are marked in red. Our point-by-point responses are as follows.

Point 1Page 1_Line 34: I think the citation must be numbered!!! Please check the journal requirements for the citation & references!!!

Point 2Move figure information after the figures!!!

Our response 1, 2: Thanks very much for the valuable comments. The manuscript has been revised according to the template of the journal. Citation & references have been renumbered. Meanwhile, figure captions have been placed at the bottom of the figures. Please kindly see the revised manuscript.

Point 3Regarding Figure 1: No need to write “published”. Remove it

Point 4: And, include “in Chinese and English” after “The number of articles”...

Point 5: Include “of articles in Chinese and English” after “cumulative numbers”…

Point 6: Close to the graph.

Our response 3-6: Thanks very much for the specific comments. There are many imprecise expressions in Figure 1, so that we recreated the Figure 1. Apart from that, we replaced “the number of published articles” with “the number of articles in Chinese and English”, and the “Cumulative numbers” with “Cumulative numbers of articles in Chinese and English”, and close to the graph as suggested (line 120). Please kindly see the revised manuscript.

Point 7: It is interesting to note that the number of Chinese articles increased between 2014 and 2018 compared to English articles. Then, this trend started decreasing for the Chinese. What was the reason for that? Explanation...

Our response 7: This is indeed a very interesting phenomenon. On reflection, we think there are three possible reasons.

  • The articles in 2021 are incomplete statistics and the time ended in September 2021, leading to a decrease in both Chinese and English articles. In fact, the full statistics for 2019 and 2020 are essentially the same as for 2018.
  • Increasing numbers of Chinese researchers tend to submit their articles to English journals. In the process of browsing relevant articles, we found that many English articles take China as the research area, but they are counted in the abroad articles.
  • Sponge city is often used when some Chinese researchers publish Chinese journals, but it covers a wider range of contents than low-impact development. In order to make the database more accurate, some articles that replace the concept of LID with the concept of sponge city were not included in the statistical analysis.

Point 8: No explanation for Table 6 in the text!!!

Point 9: Where is Table 6 in the text???

Our response 89: Thanks very much for the valuable comments. Table 6 shows the highly cited literature in LID research. In the process of revising the manuscript based on the journal template, we put it in the correct position and explained it in the manuscript (line 243). Please kindly see the revised manuscript.

Point 10: Figure 6: Please translate “Chinese words” to English in Figure 6.

Our response 10: We unanimously agree with the reviewer’s comment of replacing the Chinese parts of Figures 6 with English, which can better help readers understand the meaning of the manuscript. We recreated Figures 6 and replaced it with Table 9 (line 264) as suggested. In addition, we recreated Figures 7 and replaced it with Figure 5 (line 279). Please kindly see the revised manuscript.

Point 11: Finally, there are some mistakes in the writing of the paper.

Our response 11: Thank you for giving the suggestions. The manuscript has been remarkably improved in grammar and style by a professional English editor. Please kindly see the revised manuscript.

Point 12: Please check the attached file for my corrections/comments.

Our response 12: We appreciate you reading our article so carefully and giving so many valuable comments. We have seriously read your comments and made corrections point by point. Besides, we also noticed that you have put forward a lot of suggestions on the abstract section. A good abstract is of great necessity, so that we re-wrote the abstract section to clarify the research purpose of the manuscript, the research process and methods, and the conclusion of the research (lines 15-36). Please kindly see the revised manuscript.

Reviewer 3 Report

The present review article compares the low-impact development research articles in China and abroad regarding the urban construction model for managing floods. The manuscript is well structured. However, the main findings are not present in the abstract, which is also not indicative of the subject analyzed in the introduction and seems best suited for the materials and methods section. Some syntax and spelling errors need the attention of the authors. It's also not clear why figure 7 is presented in Chinese and the same applies to the keywords of fig 6. Moreover, the significance of the study is also not clear, or how it could be used or referenced by other scientists. The list of references is poor for a review article. 

Author Response

Dear Reviewer:

We would like to thank you for spending so much time reading our manuscript entitled “A systematic bibliometric review of low impact development research articles” (ID: water-1846275) and for providing valuable comments and suggestions. We would like to thank the reviewers for their thoughtful review of this manuscript. The comments and suggestions are very helpful for improving the manuscript. We have revised our manuscript accordingly, and the major revisions are marked in red. Our point-by-point responses are as follows.

Point 1However, the main findings are not present in the abstract, which is also not indicative of the subject analyzed in the introduction and seems best suited for the materials and methods section.

Our response 1: Thanks very much for the valuable comments. We re-wrote the abstract section to clarify the research purpose of the manuscript, the research process and methods, and the conclusion of the research (lines 15-36). Please kindly see the revised manuscript.

Point 2Some syntax and spelling errors need the attention of the authors.

Our response 2: Thank you for suggestions on syntax and spelling. The manuscript has been remarkably improved in grammar and style by a professional English editor. Please kindly see the revised manuscript.

Point 3It's also not clear why figure 7 is presented in Chinese and the same applies to the keywords of fig 6.

Our respond 3: We unanimously agree with the reviewer’s comment of replacing the Chinese parts of Figures 6 and 7 with English, which can better help readers understand the meaning of the manuscript. While recreating Figures 6 and 7, we simultaneously replaced Figure 6 with Table 9 (line 264), and Figure 7 with Figure 5 (line 279) as suggested. Please kindly see the revised manuscript.

Point 4Moreover, the significance of the study is also not clear, or how it could be used or referenced by other scientists.

Our response 4: This is a very meaningful comment, and we thought about it in detail. In this manuscript, we used Citespace software to analyze 2013 LID related literatures at China and abroad, in order to sort out and summarize related research through the knowledge map, and objectively present the subject structure, research focus and research hotspot of LID research. The analyses presented herein may be beneficial to different stakeholders in the LID research and decision-making community, including researchers (e.g., to consider breakthrough and seminal work opportunities at the interface with LID fields), designers/regulators (e.g., to improve LID design, optimization, economy and resilience), and funding agencies (e.g., to build LID knowledge architecture quickly).

Point 5The list of references is poor for a review article.

Our response 5: Thanks very much for the specific comments. We read a large number of references again, and selected some important and representative references into the manuscript (lines 357-462). Please kindly see the revised manuscript.

Reviewer 4 Report

The review paper entitled ''Correspondence: systematic bibliometric review of low impact development research articles'' reported a bibliometric analysis of the studies on low-impact development. The results presented are interesting, and the manuscript is well written but requires some minor revisions. My comments are as follows.

1.      The author should highlight the contribution of this paper to the literature.

2.      The main aim should be stated in the abstract.

3.      Line 129: a typo (froms) needs to be fixed.

4.      Figure captions should be at the bottom of the figures.

5.      Figure 6 and Figure 7: The information on the figure should be given in English.

6.      Conclusions should be improved. A conclusion on the comparison of the LID research in China and abroad should be given.

Author Response

Dear Reviewer:

We would like to thank you for spending so much time reading our manuscript entitled “A systematic bibliometric review of low impact development research articles” (ID: water-1846275) and for providing valuable comments and suggestions. We would like to thank the reviewers for their thoughtful review of this manuscript. The comments and suggestions are very helpful for improving the manuscript. We have revised our manuscript accordingly, and the major revisions are marked in red. Our point-by-point responses are as follows.

Point 1: The author should highlight the contribution of this paper to the literature.

Our response 1: Thanks very much for the valuable comments. We used Citespace software to analyze 2013 LID related literatures at China and abroad, in order to sort out and summarize related research via the knowledge map, and objectively present the subject structure, research focus and research hotspot of LID research. At the same time, these are highlighted in the abstract. Please kindly see the revised manuscript.

Point 2The main aim should be stated in the abstract.

Our response 2: Thanks very much for the valuable comments. We re-wrote the abstract section to clarify the research purpose of the manuscript, the research process and methods, and the conclusion of the research (lines 15-36). Please kindly see the revised manuscript.

Point 3Line 129: a typo (froms) needs to be fixed.

Our response 3: Thank you very much for finding this syntax error. We replaced “froms” with “from” as suggested (line 135). Furthermore, the manuscript has been remarkably improved in grammar and style by a professional English editor. Please kindly see the revised manuscript.

Point 4Figure captions should be at the bottom of the figures.

Our response 4: Thanks very much for the valuable comments. The manuscript has been revised according to the journal template, and figure captions have been placed at the bottom of the figures. Please kindly see the revised manuscript.

Point 5Figure 6 and Figure 7: The information on the figure should be given in English.

Our respond 5: We unanimously agree with the reviewer’s comment of replacing the Chinese parts of Figures 6 and 7 with English, which can better help readers understand the meaning of the manuscript. While recreating Figures 6 and 7, we simultaneously replaced Figure 6 with Table 9 (line 264), and Figure 7 with Figure 5 (line 279) as suggested. Please kindly see the revised manuscript.

Point 6Conclusions should be improved. A conclusion on the comparison of the LID research in China and abroad should be given.

Our response 6: Thanks very much for the specific comment, which helps us to improve the conclusion of the manuscript. In section 6.1, we compared the difference of LID research between China and abroad (lines 322-343). Please kindly see the revised manuscript.

Round 2

Reviewer 2 Report

I see that the authors have successfully revised the manuscript based on my comments.

Now, the manuscript can be considered for publication in the Journal of Water.